# Application of Least-Cost Movement Modeling in Planning Wildlife Mitigation Measures along Transport Corridors: Case Study of Forests and Moose in Lithuania

**Jack Wierzchowski [1], Andrius Kučas [2]**  **and Linas Balčiauskas [2,\***

1   Geoprofis Austria, Stubenerstraße 73, 7434 Stuben, Austria; jack.wierzchowski@gmail.com
2   Laboratory of Mammalian Ecology, Nature Research Centre, Akademijos 2, 08412 Vilnius, Lithuania; kucas.andrius@gmail.com
*   Correspondence: linas.balciauskas@gamtc.lt; Tel.: +370-685-34141

**Abstract:** The present work presents the development of a moose movement model to explore the value of wildlife mitigation structures and examine how hypothetical changes in land use patterns could alter wildlife habitats at landscape scales. Collisions between vehicles and animals pose a threat to humans and wildlife populations, the most dangerous collisions being with moose. Migrations of moose are generally predictable and habitat-dependent. Here, we use GIS-based simulations of moose movements to examine road-related habitat fragmentation around the main highways A1 and A2 in Lithuania. From forest data, we develop a moose habitat suitability map. Then, by running multiple simulation iterations, we generate potential moose pathways and statistically describe the most efficient potential long-range movement routes that are based on the principles of habitat utilization. Reflecting the probabilities of cross-highway moose movement, ranks are assigned to all 1 km highway segments, characterizing them in terms of their likelihood of moose movement, and thus identifying discrete migration corridors and highway crossing zones. Bottlenecks are identified through simulation, such as where sections of wildlife fencing end without highway crossing structures, thereby creating a 'spillover' effect, i.e., moose moving parallel to the highway, then crossing. The tested model has proven the prognostic capacity of the tool to foresee locations of moose-vehicle collisions with high accuracy, thus allowing it to be a valuable addition to the toolbox of highway planners.

**Keywords:** moose migrations; habitat suitability; forest fragmentation; roadkills

## 1. Introduction

The detrimental impacts of highways and transport corridors on nature conservation are well documented [1–3]. In an increasing number of environments, including forests [4], animal movements necessitate the crossing of roads [5]. With the number of roads and vehicles increasing [6], the rate that animals manage to successfully cross roads is decreasing, becoming the leading cause of animal mortality in some cases [2,7–9]. Roads and transport corridors also contribute to habitat fragmentation and the isolation of populations [10].

In road planning, authorities have generally restricted their consideration to simple one-dimensional, linear zones along roads and highways [11,12]. The planning of roads in forested areas is problematic [13]. Forests are primary habitats for many wild ungulate species. In the case of Lithuania, forests are important for moose (*Alces alces* Linnaeus, 1758), red deer (*Cervus elaphus* Linnaeus, 1758), roe deer (*Capreolus capreolus* Linnaeus, 1758) and wild boar (*Sus scrofa* Linnaeus,

1758). In recent years, collisions with European bison (*Bison bonasus* Linnaeus, 1758) and fallow deer (*Dama dama* Linnaeus, 1758) have also taken place, both of these species being forest dwellers. Compounding the issue, moose move between winter and summer grounds twice a year, thus increasing the number of moose-vehicle collisions (MVC).

However, the ecological effects of roads are often many times wider than the road itself and can be immense and pervasive [10,14]. In Norway, there are about 2000 MVC per year [15], while the number in Sweden is about 4500 per year, with 10–15 human fatalities [16]. In North America, the number of MVC in Maine exceeded 8000 between 1992 and 2005, with a yearly average of over 500 MVC [17], while in British Columbia 300 to 1200 MVC were reported annually, with one to two human fatalities per year [18,19]. In Lithuania, the number of MVC is rising every year, increasing from 12 in 2003 to 203 in 2016, this being the main cause of human fatalities in vehicle-wildlife collisions [20].

The construction of roads and associated wildlife safety measures, such as wildlife fencing, results in forest fragmentation and the genetic subdivision of moose populations [21,22]. Due to the broad landscape context of road systems, it is essential to incorporate landscape patterns and processes into the planning and construction process these systems [23]. GIS tools and applications are commonly used to accumulate pertinent spatial information, as well as to model the impacts of roads and to map habitat linkages across roads [24,25]. Wildlife movement simulation (WMS) models are a natural extension to these tools and are now being applied in the context of wildlife management [3,26].

Least-cost modeling is used to estimate the connectivity of landscape matrices [27–30]. The models assume that wildlife movement adheres to the 'least-cost' principle [31,32], which states that dispersing wildlife is most likely to follow a path of least resistance along a vector defined by the juxtaposition of high quality habitat patches, weighted by distance considerations, further modified by the presence and geometry of barriers to movement, such as fencing and contiguous urban areas [33]. In the case of moose, the main habitats are forest [34] and wooded wetlands [35]. Agricultural areas are avoided or used inconsistently [35–37].

Least-cost paths are travel routes between two given locations that incur the lowest "cost" of transit [24,32]. Within a grid of cells that make up a study area, each cell has a cost value associated with it. In many cases, least-cost path models have been shown to effectively predict road crossing corridors for carnivores [38] and other wildlife [39,40]. For example, least-cost modelling has identified wildlife corridors for fishers (*Martes pennanti* Erxleben, 1777) and bobcats (*Lynx rufus* Schreber, 1777), but has not worked well in the case of the American black bear (*Ursus americanus* Pallas, 1780) [38]. However, in other studies, the movement corridors of the African forest elephant (*Loxodonta africana* Blumenbach, 1797), forest buffalo (*Syncerus caffer nanus* Boddaert, 1785), western lowland gorilla (*Gorilla gorilla gorilla* Savage and Wyman, 1847), and central chimpanzee (*Pan troglodytes troglodytes* Blumenbach, 1775) only partially overlapped with the pathways from least-cost models [40].

This paper describes the application of a least-cost WMS model for moose, with the view of discussing recommendations for the placement of highway mitigation measures that would reduce MVCs and help maintain moose habitat connectivity along the two main roadways in Lithuania. The described method of mapping moose movement does not seek to mimic individual moose routes across a landscape, which are often driven by stochastic events that are impossible to express quantitatively. Here, we describe statistically the most efficient long-range potential movement routes based on principles of habitat utilization. Discrete migration corridors and highway crossing zones were highlighted and ranked. Model testing has proven the prognostic capacity to foresee locations of moose-vehicle collisions with high accuracy, thus showing this model to be a valuable addition to the toolbox of highway planners.

## 2. Material and Methods

### 2.1. Study Area

In 2014, we simulated moose movements for the 311 km A1 highway connecting Vilnius with Klaipėda, and the 136 km A2 highway linking Vilnius with Panevėžys. Highways A1 and A2 are the

busiest roadways in Lithuania, with traffic figures of 16,873 and 9523 vehicles/day, respectively, in 2014. Here, the study area consisted of a 30 km wide buffer on each side of the road along the entire length of both highways (Figure 1A). The buffer area, amounting to 26,429.38 sq. km, covers 40.7% of the territory of Lithuania. Forest cover in the buffer totals 8856.48 sq. km, i.e., 33.5%, which is approximately the same as the figure for the average 33.1% forest cover in Lithuania as a whole.

    Moose movements across highways A1 and A2 were affected by wildlife fencing, forcing moose to move along the fences and permitting road crossing only where underpasses and river bridges with sufficient dimensions existed (Figure 1B). The use of existing underpasses by moose was confirmed in 2014–2015 by camera traps (Balčiauskas et al., unpubl.).

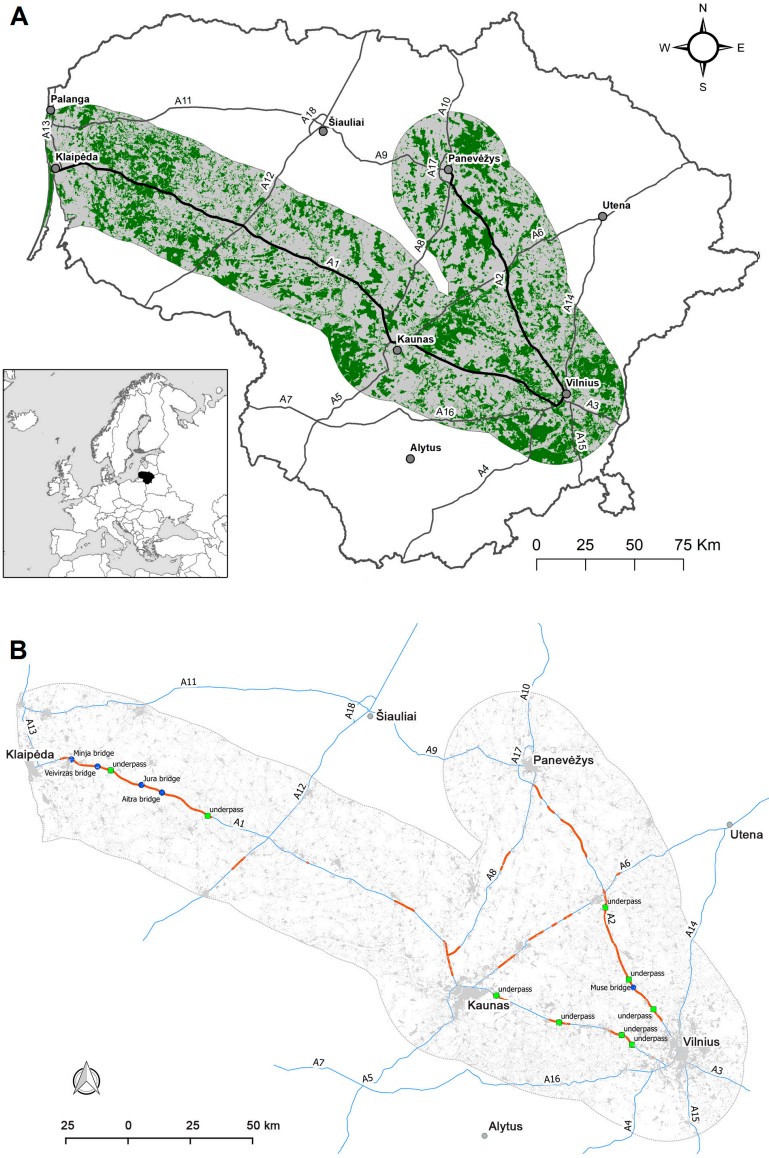

**Figure 1.** Study area: Map of highways A1 and A2 in Lithuania (**A**): The shaded area shows the 30 km buffer either side of the highways. The green color represents forests. Wildlife highway fencing and crossing structures on the highways within the study area, as used in the "current fencing" model are shown, including (**B**) fencing (red lines), underpasses (green squares), and bridges (blue dots), which are all as of 2013. Fencing on the A16 is not shown. River bridges with dimensions (span and height) that might allow cross-highway movement of moose are also shown. The map was created in QGIS, version 2.8 "Wien" (http://www.qgis.org/), licensed under the GNU General Public License, using background data from the Lithuanian Asset Information System (http://lakis.lakd.lt/lakis/).

### 2.2. Moose-Vehicle Collision Data

MVC data were obtained from the Lithuanian Police Traffic Supervision Service (TSS) and collected by the Nature Research Centre (NRC). The TSS data cover the period of 2002–2017, while the NRC data were collected by professional theriologists over the period of 2007–2017. MVC locations were registered with a precision of 100 m (or GPS coordinates), which were later converted to a GIS layer.

An MVC sample of $n = 399$ in 2002–2014 on highways A1 and A2 was used to calculate their median distances to houses and other dwellings, while a sub-sample with $n = 59$ MVC (32 on A1 and 27 on A2) that occurred in 2009–2013 was used for validating the moose habitat model. A sample with $n = 78$ MVC from 2015–2017 was used to see if the model still worked.

### 2.3. Workflow

In order to describe moose movement based on the least cost principle, the following workflow was used:

- Selection of the most suitable model and creation of the moose habitat map;
- Adaptation of the selected model to the study area if required;
- Checking if habitats other than forests should be included on the map;
- Creation of the "potential" moose habitat map based on the habitat suitability index (HSI) values, and a "realized" habitat map, imposing the negative impact of human activity;
- Validation of the created moose habitat map (comparing actual moose densities to habitat suitability according to the model at the municipality level);
- Simulation of moose movements according to the habitat (using least cost principle and values of habitat suitability). The simulation was run twice, the first time based only on the realized habitat, the second one including wildlife fences as impermeable barriers to moose movement. As a result, moose movement pathways were obtained based on suitable habitats, and locations of the most probable moose crossing localities were defined;
- Validation/testing of the moose movement model (comparing the locations of actual MVC with the predicted zones of moose road crossing).

### 2.4. Model selection

We developed a digital habitat map for our study area based on a review of pertinent literature on moose ecology and moose habitat selection. We reviewed over 50 publications from the Web of Knowledge and concluded that the best argued and documented was the moose habitat model developed for the Lake Superior region in the USA [41,42]. This model has been applied elsewhere, for example, in the state of New York, USA [43], and in Fennoscandia for modeling habitat suitability in coastal regions [44].

The Lake Superior region shares similar physiography and climate with Lithuania. The vegetation cover is also similar in both areas, with forest areas dominated by coniferous stands, namely spruce (*Picea* sp.), fir (*Abies* sp.) and pine (*Pinus* sp.), and considerable areas also of deciduous forests, namely birch (*Betula* sp.), poplar (*Populus* sp.), alder (*Alnus* sp.), and willow (*Salix* sp.). Both areas have a significant share of riparian, lacustrine, riverine and palustrine environments, and a similar relatively gentle topography. The major difference between the regions is the level of fragmentation of natural habitats, which is greater in Lithuania, where forests cover about 33% of the country [45].

### 2.5. Adaptation of Allen's Model II for Lithuania

We chose Allen's model II [41] to build a year-round moose habitat map for our study area, as it allows the evaluation of moose habitat based solely on vegetative (mainly forest) cover. In our study, we used detailed 1:10,000 digital forestry maps provided by the Nature Research Centre, Lithuania. With these maps, information on forest composition and age is present, listing dominant and other tree species, with other habitats (shrubs, wetlands, open areas, clear-cuts, etc.) presented at the

parcel (smallest uniform forest area) level. The digitized maps are based on former paper maps of the forest cadastre, while ground truth was conducted for all forested territory of the country and is constantly renewed.

Model II is based on the assumption that in the absence of critical mortality factors, moose populations with an abundance of quality food and cover have the potential to increase or stabilize at relatively high densities [41]. Lower HSI values indicate a capacity to support correspondingly fewer moose.

The Allen's model II habitat suitability index was calculated as:

$$\text{HSI} = (\text{S1} \times \text{S2} \times \text{S3} \times \text{S4})^{1/4} \tag{1}$$

where S1 to S4 represent the below-mentioned cover types. These variables are assumed to have equal weight in deriving the composite HSI value. For HSI >0, all four habitat components must be present within the evaluation area.

The model assumes that the optimal composition of cover type composition within an evaluation unit is achieved when:

- 40% to 50% of the evaluation area is covered by sites with at least 50% of shrubs or forest <20 years old (S1); these early succession stages are assumed to provide abundant, preferred forage for moose;
- 5% to 15% of the evaluation area is dominated by spruce/fir over 20 years old (S2), providing optimal availability of winter cover for moose;
- 35% to 55% of the evaluation area is dominated by upland deciduous or mixed forest ≥20 years old (S3). These forest cover types are assumed to provide food as well as cover;
- 5% to 10% of the evaluation area is wetlands dominated by open water, emergent vegetation or submerged/floating-leaved hydrophytes (S4) [41].

We, however, did not use a quantitative habitat map as the basis for performing moose movement simulations, as we were not trying to assess the moose carrying capacity of the study area. In this context, the method of calculating the composite HSI index, as presented in Equation (1), is far too restrictive. We presumed that the absence of any of the model components (S1 to S4) in an evaluation area should not to reduce its value to zero, but instead reduce it proportionally. We decided to use an arithmetic mean to compute the composite index and we believe that our less restrictive equation (2) better represents the value of the evaluation areas for migrating or foraging moose:

$$HSI = (S1 + S2 + S3 + S4)/4 \tag{2}$$

Model II is intended to be applied to large evaluation areas (minimum 9000 ha) that can support entire moose populations over a reasonably long-term basis [41], however, we were interested in doing the evaluation at the level of an individual moose. Therefore, we reduced the size of the evaluation areas to 600 ha, a size comparable to the area of a moose home range. Allen [41] stated that "600 hectares can potentially provide the annual habitat requirements of moose". This size falls within known moose home ranges [46]. Despite the reduction of scale, the moose habitat requirements and the interpretation expressed by Allen's HSI index should still hold true.

In our calculations, we used a moving window technique with a circular kernel with a 1380 m radius that yielded a circle having an area equal to 600 ha. The kernel was moved across the study area with a step equal to the GIS layers' resolution (50 m), and cover type compositions for each of the Model II variables were calculated (S1 to S4).

We analyzed MVC locations from 2002–2014 ($n = 399$) to calculate their median distances to houses and other dwellings based on the digital 1:10,000 topographic maps of the country (https://www.geoportal.lt/geoportal/nacionaline-zemes-tarnyba-prie-zemes-ukio-ministerijos). The median distance to buildings was 316 m (Figure 2A). We found that the simple equation ($y = 0.0033x$)

expressed a linear reduction of complete (100%) habitat loss immediately adjacent to the built footprint, to no reduction whatsoever at a distance of 300 m from them. The potential habitat maps were adjusted accordingly.

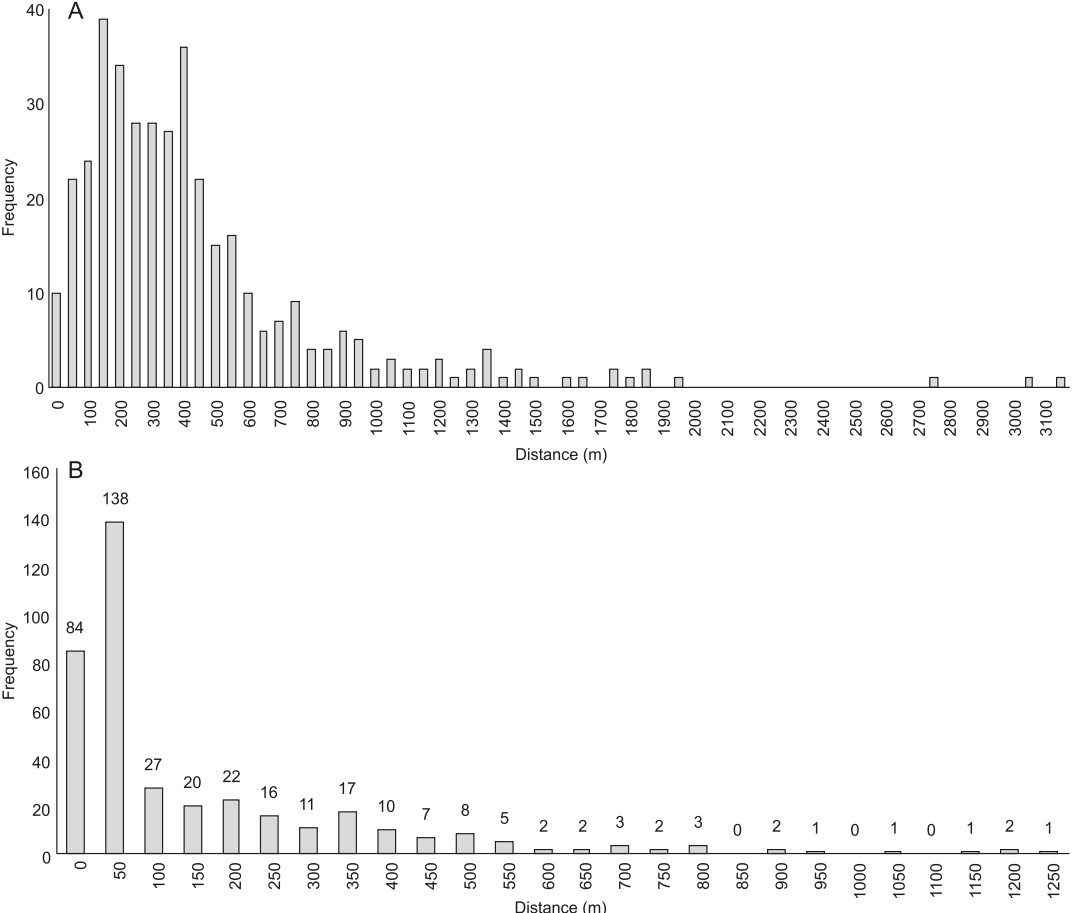

**Figure 2.** Histograms of distances from moose-vehicle collisions (MVCs) against the built-up footprint (**A**) and forest cover (**B**) in Lithuania. Digital 1:10,000 topographic maps of the country (https://www.geoportal.lt/geoportal/nacionaline-zemes-tarnyba-prie-zemes-ukio-ministerijos) were used to define buildings/dwellings footprints.

The median distance of MVC from forest (i.e., hiding) cover was 35 m, while 57% and 70% of MVC occurred closer than 50 m and 150 m to forest areas respectively (Figure 2B). Thus, MVC seldom occurred in areas of open fields. Given these uncertainties, we decided not to consider agricultural areas as either negatively or positively influencing moose movements across the landscape.

### 2.6. Validation of the Moose Habitat Model

Several methods can be used for validating habitat models, among them moose density information and habitat use [42]. Within our study area, we compared normalized average HSI values to normalized moose density averages. Normalization using the formula (value − min)/(max − min) ranged all values from 0 to 1.

Normalized HSI values were calculated for the 23 (out of 60, with range from 440 to 2216 km$^2$ and mean = 1282.2 km$^2$; https://osp.stat.gov.lt/regionine-statistika-pagal-statistikos-sritis) national municipalities (or their parts) within the 30 km buffer around both highways. Normalized 5-year (2009–2013) moose density averages were calculated from official moose survey data (data sourced from the Ministry of Environment of the Republic of Lithuania) in the municipalities of Lithuania.

We also validated our HSI model by calculating the distance of MVC locations in 2009–2013 (*n* = 59) on the two most intensively used highways to the areas having above-average HSI values, comparing them to the distances calculated for the randomly generated locations (*n* = 59) on highways A1 and A2.

*2.7. Moose Movement Simulation Model*

In our WMS model, cost can be thought of as an index of habitat quality that expresses food availability and security, where a lower cost of movement is associated with a high quality habitat and low levels of human disturbance, while high costs are associated with a poor habitat and high levels of human disturbance. If travel occurs parallel to a cell side, the cost of travel equals the value of the cell multiplied by 1.0. If travel occurs diagonally across a cell, the cost of travel equals the value of the cell multiplied by 1.414 (the length of the diagonal of the cell). The cost of an entire route is the accumulated cost of all cells along the route. The 'cost' layers were produced using the "COSTGROW" algorithm provided in the IDRISI GIS software (version Kilimanjaro, Clark Labs, Clark University, Worcester, MA, USA), which calculates a cumulative cost of movement from a specified location [47]. Least cost pathways were generated by an algorithm that seeks out minimum values on the cost surface, starting from a target location (exit point) and finishing at a source location (entry point).

We used our Allen's model II HSI map as the basis for the least cost modeling. Only one of four model II variables (S3) did not explicitly or solely relate to moose feeding requirements. Given that moose must ingest 30+ kg wet-weight of browse during the 5 to 8 h spent feeding per day, sources of forage must be concentrated [41]. Therefore, we converted our HSI map into a friction surface using the empirically derived formula that ensured that the generated pathways closely followed high quality habitat patches expressing high theoretical forage potential:

$$\text{When HSI } = \text{ 0.00–0.40, friction value } y = 5.12 \times e^{-6.9315 \times HSI} \tag{3}$$

$$\text{When HSI } = \text{ 0.41–0.50, } y = -2.88 \times HSI + 1.472 \tag{4}$$

$$\text{When HSI } = \text{ 0.51–1.00, } y = 1.024 \times e^{-6.9315 \times HSI} \tag{5}$$

When simulating wildlife movements based on the least-cost principle, the derivation of such equations (3,4,5) or defining movement resistance values is of paramount importance [24,27,32,48]. The method of summation of costs in the COSTGROW algorithm implies that the shortest path tends to have the lowest accumulated cost of movement. We used the algorithm that maximized median HSI values summed up along the entire pathway length, while providing for the shortest possible route connecting any given entry and exit points (Figure 3A).

Friction grids were further modified to include known barriers to wildlife movement. We used 1:10,000 digital topographic maps of the study area to extract human land use features that were deemed uncrossable barriers to moose movement, such as contiguous built-up areas or industrial sites.

We simulated moose movements across highways A1 and A2 using nearly 200 entry and exit points located on both sides of the highways. We generated the locations randomly, imposing a restriction that they fell within moose habitat patches with HSI values >0.6, however, to reflect the best available habitat. The number of entry/exit points was further stratified to reflect moose densities reported for the municipalities. Since we were interested in modeling mostly long-range movements, entry/exit points were further restricted to habitat patches at least 6 km away from the highways. The distance is related to the minimum area that can support the long-term existence of the moose population [41].

For each simulation, five pathway iterations were calculated. The first iteration simulated the least-cost movement with no obstructions imposed. In the second iteration, the footprint of the first pathway was assigned a high friction value, forcing the creation of a new route, distinct from the original. In the third iteration, the first two pathways were also "blocked", and an alternative route

was generated, and so on (Figure 3B). This multiple iteration approach allowed a broad spectrum of potential movement pathways to be generated and statistically characterized. For each pathway, the model calculated the path's level of complexity (a measure of how a pathway diverges from a straight line), a median of the crossed moose habitat and the median distances to forest cover and to human development. The number of pathway/asphalt road crossings was also computed.

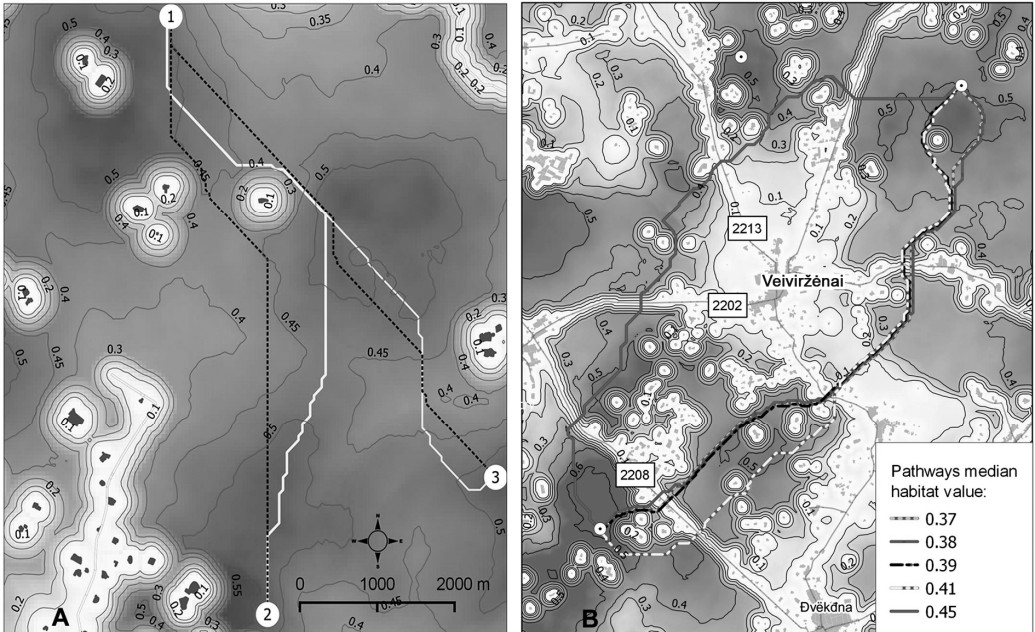

**Figure 3.** Example of pathways generated by using different methods of (**A**) the habitat suitability index (HSI) to friction transformations and (**B**) pathway iterations generating distinct movement routes between given entry-exit pairs on the optimized friction surface. In (a), black dashed lines show the pathways generated using a simple friction layer obtained by reversing the HSI values (1 minus HSI). Solid white lines show the pathways generated by applying our equations to calculated friction values. Note that this path follows high HSI values much more closely. The grey backdrop shows the moose HSI map (darker grey representing a higher HSI). Numbers in rectangles represent asphalt road numbers. The map was created in QGIS version 2.8 "Wien" (http://www.qgis.org/), licensed under the GNU General Public License, using background data from the Lithuanian Asset Information system (http://lakis.lakd.lt/lakis/).

Next, each highway was divided into 1 km segments. For each segment, the number of pathways crossing it and their cumulative statistics (median of potential and realized habitat quality, distance to human land use, pathway complexity, and proportion of pathway within forest cover) were calculated. These calculations were carried out for the entire pathway lengths and for pathway sections within a 500 m wide buffer on either side of the highways. Based on expert opinion, each statistic was then assigned a weight and combined linearly (weighted average) to rank each highway segment in terms of it being a probable moose crossing zone.

We used Saaty's method [49] to generate weights, which produced weights by means of the principal eigenvector of a pair-wise comparison matrix. This procedure generated an internally consistent set of weights and produced an index (consistency ratio) that estimated the probability that the weights were not assigned randomly (Table 1).

We computed three crossing zone ranks, the first of which is rank 1, which uses the potential moose habitat map. Rank 2 uses the realized habitat map (Figure 4), which imposes human impact, reducing habitat availability. Rank 3 was used for the 500 m buffer around the highways. For a

more transparent characterization of the crossing zones, and to reflect variability as well as spatial distribution of these habitats, we combined ranks 1, 2, and 3 into a cumulative value (Equation (6)):

$$Rcumulative = (R1 + R2 + 2 \times R3)/4 \qquad (6)$$

**Table 1.** Eigenvector (weights) analysis for moose, calculated according to Saaty (1977). The highest weights were given to the habitat and hiding cover layers. Consistency ratio = 0.05 (full pathway lengths) and 0.001 (500 m buffer).

| Full Length of Pathways | | Area within 500 m Buffer Each Side of Highways | |
| --- | --- | --- | --- |
| **Layer** | **Weight** | **Layer** | **Weight** |
| Moose habitat | 0.3787 | Moose habitat, side A | 0.2940 |
| Path proportion in hiding cover | 0.3645 | Moose habitat, side B | 0.2940 |
| Distance to human development | 0.1516 | Distance to human development, side A | 0.0588 |
| Number of roads crossed | 0.0771 | Distance to human development, side B | 0.0588 |
| Pathway complexity | 0.0281 | Path proportion in hiding cover | 0.2941 |

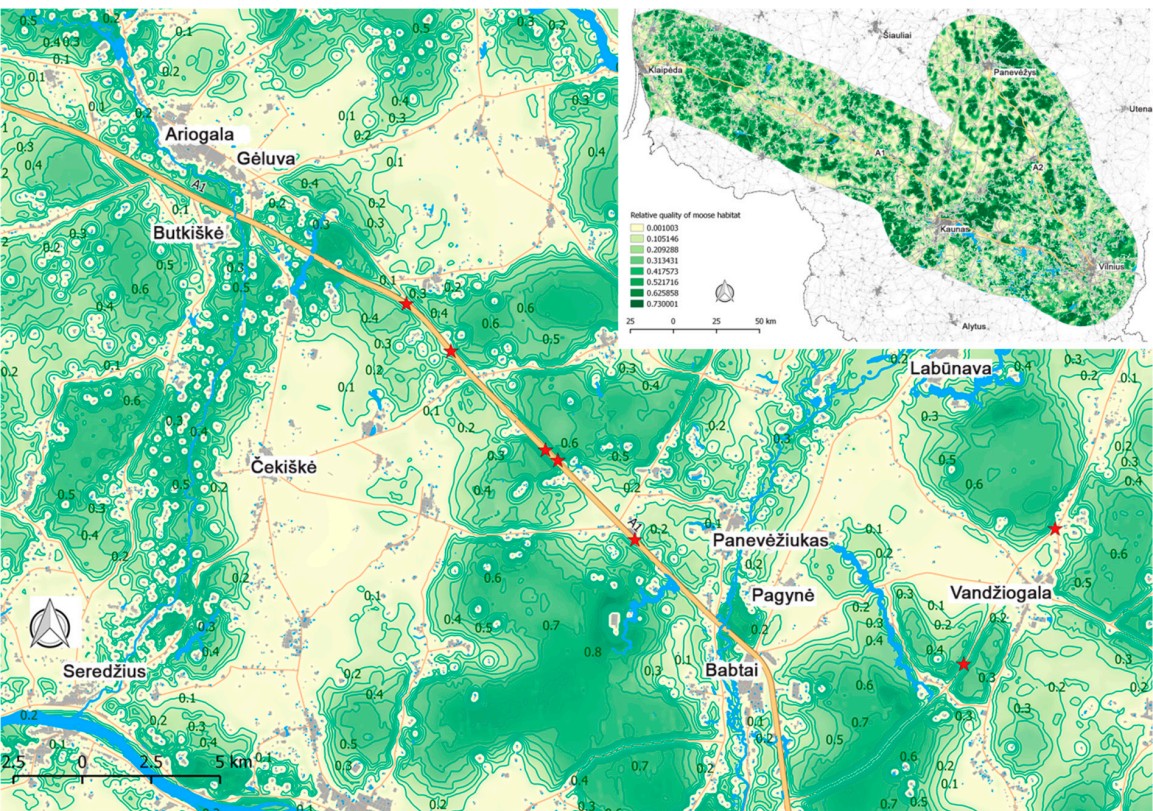

**Figure 4.** Close-up of the realized moose habitat map with HSI contour lines. Stars mark the locations of moose-vehicle collisions from 2002–2013. The realized habitat map for moose within the 30 km buffer each side of highway A1 and A2 is shown as an inset. The map was created in QGIS version 2.8 "Wien" (http://www.qgis.org/), licensed under the GNU General Public License, using background data from the Lithuanian Asset Information System (http://lakis.lakd.lt/lakis/).

Two simulations, the first assuming no fencing along the highways, the second with all currently-built fencing considered as an absolute barrier to movement (see Figure 1), produced 3795 distinct moose movement pathways. Data on existing wildlife fencing and wildlife underpasses (as of 2013) for all the highways within our study area (provided by the Lithuanian Road Administration) were used. We used the output of both models to draw recommendations regarding the placement of

further highway mitigation measures. The 'no fencing' model was used to identify habitat-driven optimal crossing zones, while the 'current fencing' model helped to identify those sections of the highways where a combination of the existing wildlife fencing and a lack of highway crossing structures created the potential for a spillover effect, i.e., moose moving parallel to the highway and crossing at the end points of the fences.

### 2.8. Moose Movement Model Testing

We tested our model's predictive powers by statistically comparing the locations of actual MVCs to the locations of all crossing zones identified by the model and to the high frequency crossing zones, as defined as highway segments that registered above-median frequencies of crossings.

We generated a random set of locations ($n$ = 32 for highway A1 and $n$ = 27 for A2, equal to the number of empirical MVC locations during 2009–2013 on these roads) and calculated the distances from both sets of points to all crossing zones and high frequency crossing zones.

We tested the following null hypothesis: $H_0$ = the distribution of MVC sites expressed in terms of distance to the simulated moose crossing zones is random.

Our alternative hypothesis stated: $H_A$ = MVC sites are closer to the simulated moose-road crossing zones than the random sample.

We used the Wilcoxon rank sum test with continuity correction.

### 2.9. Prognostic Value of the Moose Movement Model

To evaluate the prognostic capability of the model, we compared the HSI and moose movement model data from 2015–2017 MVC locations and clusters (short and significant road sections where MVC occur). Out of 78 MVC cases (Table S1), 15 were in localities not covered by the moose movement model, as these locations were within the two biggest cities where forest habitats were not present. For the remaining 63 MVC locations, we calculated the distance to the nearest place predicted by the HSI and moose movement models.

MVC data clustering was conducted using the KDE+ software package [50,51]. Finally, using standard spatial analysis tools in a GIS environment, we located the MVC hotspots and checked if they were within the fenced road sections. Cluster locations were calculated yearly for 2015, 2016, and 2017 MVC (Table 2) and for the pooled 3-year data (Table 3). Again, we calculated the distance to the nearest place predicted by the HSI and moose movement models.

## 3. Results

### 3.1. Validity of the Moose Habitat Model

The comparison of the standardized average HSI values calculated for the municipalities (or their parts) within the 30 km buffer around both highways to the standardized 5-year (2009–2013) moose density averages showed good agreement between the two data sets ($r$ = 0.7). The exceptions were four municipalities where the model overestimated the moose habitat potential (Figure S1).

The distribution of distances between MVCs and sites with above-average HSI values and between MVCs and random locations was different (Figure S2), and the median distances, being 403 m and 1344 m, respectively, differed significantly (Wilcoxon $W$ = 936, $P$ < 0.0001).

### 3.2. Moose Movement Model Results

In the 'no fencing' model, 72 highway segments (1 km long) along highway A1 registered at least one pathway crossing, of which 35 segments registered above-median (>15) crossings (Figure 5A). For highway A2, 37 segments registered one or more crossings, and 18 segments registered above-median (>21) crossings (Figure 5B). These zones of increased simulated moose traffic indicate the locations of potential moose movement zones across the highways. According to the model's assumptions, the viability of such zones is related to their cumulative ranks, where segments

that showed relatively high rank indicated not only good overall conditions for movement along the full length of the simulated paths, but also relatively good conditions within the critical 0.5 km wide zone each side of the highways, where the roadways' impact on wildlife habitat was reported to be most profound. Figure 6 shows the model output for a section of highway A1.

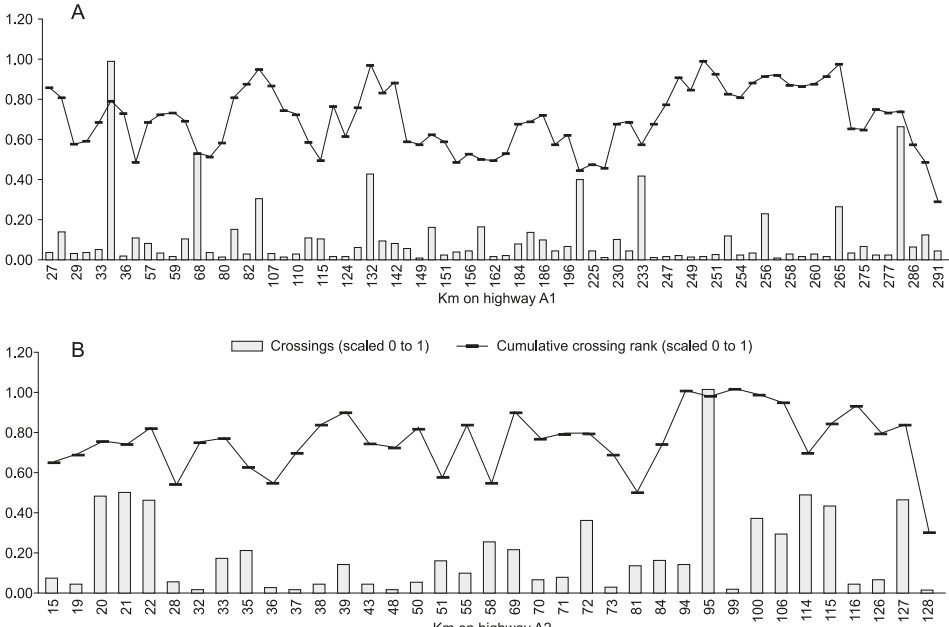

**Figure 5.** Moose crossings and their cumulative crossing ranks on 1 km segments of highways A1 (**A**) and A2 (**B**).

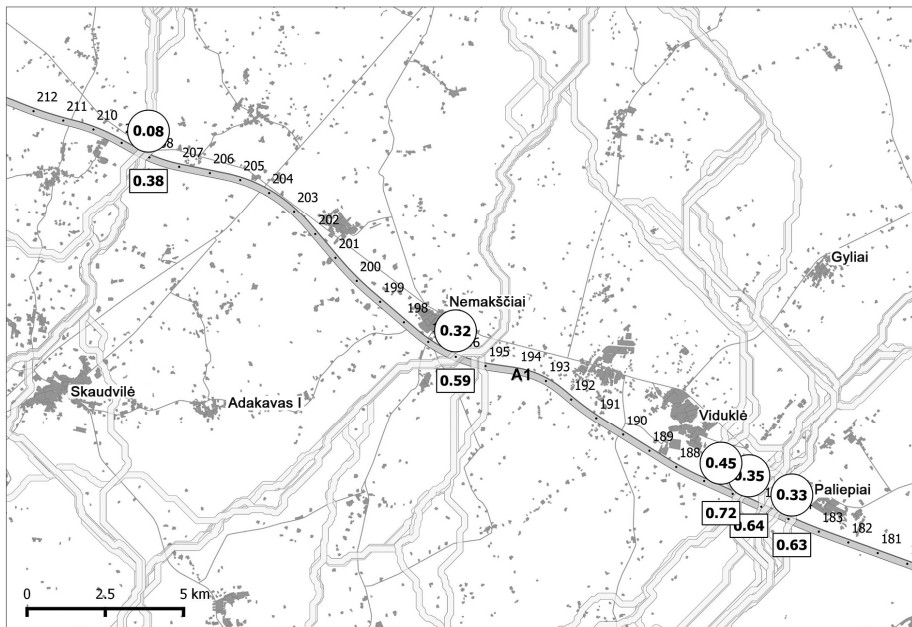

**Figure 6.** Example of model output for highway A1. Grey lines represent simulated paths. Numbers in circles above highway represent segment ranks based on conditions within a 1 km buffer on both sides of the highway. Numbers below (in rectangles) represent segment ranks based on the full pathway statistics. Numbers from 182 to 212 represent sequential 1 km long highway segments. Map was created in QGIS version 2.8 "Wien" (http://www.qgis.org/), licensed under the GNU General Public License, using background data from the Lithuanian Asset Information System (http://lakis.lakd.lt/lakis/).

In the model with wildlife fences, the number of moose crossings was predictably less. For highway A1, 33 segments with moose crossings were registered, and only 11 such segments on highway A2. Spillovers were identified at the 38, 54, 78, and 138 km markers on highway A1, and at the 22, 91, and 114 km markers on highway A2. Actual MVC data from 2002–2014 confirms the presence of a spillover effect. The most notable difference in the model with fences is the reduced number of moose crossings in the western part of highway A1 (a lack of crossings at 230, 260, and 280 km).

Pattern analysis (frequency of crossings) of simulated moose movement, combined with the analysis of the characteristics of the crossing zones allowed us to produce a set of detailed mitigation recommendations for highways A1 and A2.

### 3.3. Moose Movement Model Testing Results

Model testing revealed that for highway A1, actual MVCs were located closer to the high frequency crossing zones than to the sites of the random sample ($W = 300.5$, $P = 0.011$). The median distance from the MVCs to high frequency crossing zones was 791 m, while the median distance to random locations was 2616 m. Hence, on highway A1, high frequency crossing zones should be interpreted as likely moose crossing locations.

For highway A2, actual MVC locations were closer to all highway crossing zones identified by the model ($W = 185$, $P = 0.003$). The median distance from MVCs to high frequency crossing zones was 291 m, while the median distance from random locations was 1792 m. Hence, when analyzing the results for this highway, consideration should be equally given to all identified crossing zones.

### 3.4. Prognostic Value of the Moose Movement Model

In 2015–2017, four yearly clusters of MVC were found along highway A1 and none on highway A2 (Table 2). Three clusters were 1–2 km from the road segments, with an extremely high crossing rank, while the fourth was directly in the road segment, with an average crossing rank, and 40 crossing pathways for the 44–45 km segment.

**Table 2.** Evaluation of the prognostic possibilities of the moose movement model for highway A1, comparing with MVC clusters for every year in 2015–2017: Bandwidth 150 m, Rc—cumulative segment rank, Npath—number of pathways crossing the segment. Dist.—distance (km) between cluster and nearest Rc/Npath value.

| Year | Cluster | | | Fencing | Rc | Npath | Dist. |
| | Strength | Start | End | | | | |
| --- | --- | --- | --- | --- | --- | --- | --- |
| 2017 | 0.50 | 44.8 | 45.0 | no | 0.4 | 40 | 0 |
| 2016 | 0.67 | 135.8 | 136.0 | no | 0.82 | 35 | 2 |
| 2016 | 0.50 | 130.3 | 130.5 | no | 0.97 | 132 | < 1 |
| 2015 | 0.50 | 136.4 | 136.6 | no | 0.82 | 35 | 1.5 |

The pooled MVC data for 2015–2017 yielded five clusters for A1 highway, two of these were situated directly in the road segments with a high crossing rank (Table 3). Only one MVC cluster was found for highway A2, this being 5 km from the segment where the model shows a high crossing rank and number of pathways crossing the segment. However, the location of this cluster depended on fencing installed after 2014, so the model was not able to account for it.

**Table 3.** Evaluation of the prognostic possibilities of the moose movement model for highways A1 and A2, comparing with the MVC cumulative data for 2015–2017: Bandwidth 150 m, Rc—cumulative segment rank. Npath—number of pathways crossing the segment. Dist.—distance (km) between cluster and nearest Rc/Npath value.

| Road | Cluster | | | Fencing | Rc | Npath | Dist. |
|------|---------|-------|-----|---------|------|-------|-------|
| | **Strength** | **Start** | **End** | | | | |
| A1 | 0.67 | 135.8 | 136.0 | no | 0.82 | 35 | 2 |
| A1 | 0.50 | 44.8 | 45.0 | no | 0.4 | 40 | 0 |
| A1 | 0.50 | 136.4 | 136.6 | no | 0.82 | 35 | 1.5 |
| A1 | 0.50 | 130. 3 | 130.5 | no | 0.97 | 132 | < 1 |
| A1 | 0.50 | 265.4 | 265.6 | yes | 1.0 | 95 | 0 |
| A2 | 0.44 | 46.7 | 46.9 | yes | 0.92 | 22 | 5 |

Here, the prognostic capability of the model to indicate MVC locations was even better (Table S1). Excluding the territory of the two biggest cities located on the A1 highway, out of 63 MVC registered in 2015–2017, the location was predicted exactly in 19% of cases, while the location was predicted to the nearest km in 47.6% of cases and to an accuracy of 2 km in 66.7% of cases (Figure 7). In 25.4% of MVCs, the location was predicted to an accuracy of 3–5 km. Again however, the most pronounced deviations were in localities where new wildlife fences were installed after 2014.

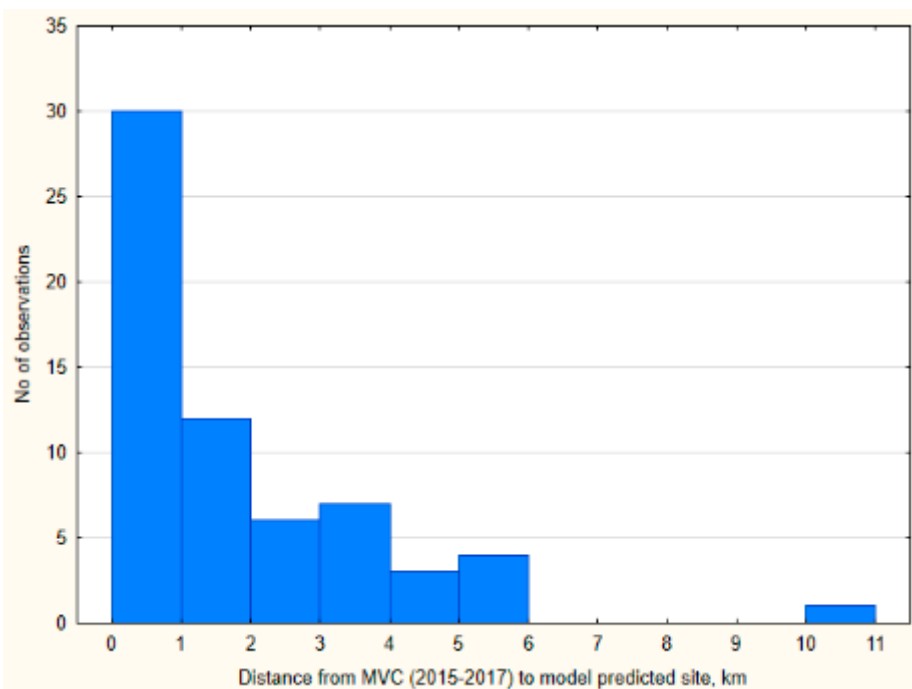

**Figure 7.** Prediction of moose movement model showing the distribution of distances between MVC locations in 2015–2017 (*n* = 63) and locations predicted by the model in 2013. Distances are rounded to the nearest km; for higher accuracy see Table S1.

## 4. Discussion

We found that the model based on the HSI and least cost movement adequately describes long range moose movements and locations of highway crossings. The generation of thousands of pathways indexed to characterize the traversed areas in terms of their ability to sustain moose movements allowed the application of statistical methods to outline and rank discrete migration corridors and highway crossing zones. Modelling without account of the existing wildlife fences along the highway produces a considerably higher number of pathways and road crossings. Introducing the fencing

into the model showed significant spillover and movement funneling effects, as well as a reduced number of crossings. The predicting ability of the model was found adequate until the installation of a considerable amount of new wildlife fencing. Thus, our model can be used in road ecology for the planning of safety measures aimed at reducing MVCs on highways. The model can explore not only the value of alternative placements of wildlife mitigation structures, but also examine hypothetical changes to land use patterns that could have the potential to alter wildlife habitat even at a considerable distance from transport corridors.

For other species of wildlife, the least cost path models identified road crossings and movement pathways with differing levels of success [38–40]. In our case, the predicting ability for highway A1 was better than highway A2. This related to the greater amount of the fencing introduced to highway A2 after the model creation. However, MVCs do occur within the fenced sections of the highways, not necessarily exactly where the moose access the road (for example, through an unlocked gate). Two MVC clusters were formed within the fenced segments in 2015–2017. On the road side of the fence, the animal may move parallel to the highway for some distance before attempting to cross. When working with low sample sizes, such MVCs are likely to skew any statistical analysis.

We modified Allen's model II (as it produces only a potential habitat suitability map), introducing the impact of human activities to generate a realized habitat layer. The majority of the reviewed literature suggested a moderate to strong negative response by moose to human development. It was found that moose showed an avoidance of areas up to 500 m from highways, and that moose crossed highways and forest roads 16 and 10 times less frequently than expected, respectively [52,53]. Others have reported the preference of moose for areas of lower road density and human development [54,55]. A recent study in Sweden reported that the average distance to roads and dwellings/buildings was different for migratory and non-migratory moose. For roads, it ranged from 131 m for non-migratory moose to 292 m for migratory moose, while for dwellings and buildings, the average distances were 331 m and 906 m, respectively [56]. Results from a study along a road corridor in Denali National Park in Alaska suggested that moose avoided areas within 300 m of the road [57]. In our study, the median distance to buildings (316 m) was similar to that reported by [56].

In the reviewed literature, we found contradictory information on the response of moose to agricultural areas and the usage of crops [37,54,55,58–60]. Even in the winter season, moose were reported to browse 95% of the time within a zone no more than 80 m from cover [61]. Our results (Figure 2) show that MVCs occur in close proximity to forest cover. Thus, for the realized habitat map, the influence of agricultural areas was not reflected.

Road crossing intensity depended on the moose population size around the highways. We suppose that, despite good agreement between average HSI values and 5-year moose density averages in the municipalities (or their parts) within the 30 km buffer on each side of both highways, the model overestimated moose habitat potential in four municipalities. In these four municipalities, however, a relatively high number of MVCs were reported [62]. Given that moose density (or moose evidence) and habitat quality is correlated with MVC locations [16,63–65], moose census numbers for these municipalities were likely to be underestimated. Insufficient knowledge of moose numbers may influence the selection of entry/exit points and, ultimately, moose movement pathway locations. We do not exclude other possible sources of bias, such as highway design and traffic signs or topographical influences [55,66].

It is important to bear in mind that the described method of mapping moose movement did not seek to mimic individual moose routes across a landscape. Such routes are often driven by stochastic events that are impossible to express quantitatively. The intent of this model was to describe statistically the most efficient potential movement routes that are based on principles of habitat utilization. The model's resolution is determined by the accuracy and representativeness of spatial data. Thus, for the implementation of the model prognostics into road ecology (i.e., the planning of fencing and under/over passes) it is advised to utilize 'on site' investigations of moose numbers and limiting factors.

We used MVC locations to test the model results and the prognostic capacity. These 'unsuccessful' highway crossings can also be thought of as a sub-sample of successful crossings occurring at the same locations. Despite tests being successful, snow tracking or telemetry data could also be used to test the model's predictive powers.

## 5. Conclusions

Here, we have developed a moose movement simulation model and identified sections on two main highways in Lithuania that needed more detailed analysis. Our model provided a spatial context for examining environmental impacts imposed by transport corridors. This was tested with moose-vehicle collision data afterwards, where the model proved to not only be accurate, but also have a prognostic capacity. Thus, it can be used for planning measures for alleviating the detrimental effects of transport corridors on habitat fragmentation, as well as for the testing of existing wildlife mitigation measures on highways, exploring alternative locations for their placements, and modelling of land use changes and their effects on road ecology. To maintain the accuracy of moose movement prognosis, we recommend to periodically re-run the model, incorporating significant changes in the built-up footprint and wildlife fencing.

**Supplementary Materials:** The following are available online at http://www.mdpi.com/1999-4907/10/10/831/s1, Figure S1: Comparison of standardized (expressed on the scale of 0 to 1) average HSI values calculated for administrative municipalities with standardized moose density values (5-year averages for 2009–2013) calculated for the municipalities, Figure S2: Line histogram comparing the frequencies of distances from MVC and random locations to habitat patches with above-average HSI values, Table S1: Evaluation of the prognostic possibilities of the moose movement model for highway A1, comparing with MVC 2015–2017.

**Author Contributions:** Conceptualization, J.W.; Methodology, J.W.; Modelling, J.W.; Formal analysis, J.W., A.K., L.B.; Data supply, L.B., A.K.; Data curation, L.B.; Cluster modelling, A.K.; Writing, J.W., L.B.; Review and editing, L.B., A.K.; Project administration, L.B.; Funding acquisition, L.B.

**Funding:** This research was funded by the Lithuanian Roads Administration of the Ministry of Transport and Communications of the Republic of Lithuania, grant No S-90.

**Acknowledgments:** Jos Stratford helped with the language. Digital landscape data were supplied by the Nature Research Centre, Lithuania.

**Conflicts of Interest:** The authors declare no conflict of interest. The funders had no role in the design of the study, in the collection, analyses, and interpretation of data, in the writing of the manuscript or in the decision to publish the results.

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
