# Peer review of "Application of Least-Cost Movement Modeling in Planning Wildlife Mitigation Measures along Transport Corridors: Case Study of Forests and Moose in Lithuania"

_forests, doi:10.3390/f10100831_

Round 1
Reviewer 1 Report
This paper examined the combination of habitat suitability models with least cost paths to determine likely areas for highway crossings of moose in Lithuania. The authors apply a previously developed habitat model from Lake Superior to the forest structure data available in their study area, and incorporate additional features such as impacts from fences and buildings. They then construct multiple iterations of least cost paths across two main highways in their study area, and discuss the implications of their findings for conservation and highway planning. I found this study interesting and likely to be a good contribution to the wildlife vehicle collision literature. I think the study as written is very unclear, however, and difficult to follow. I think substantial editing and revision, particularly in the Methods and Discussion, are needed. My specific comments are below.
Abstract Line 29: You state that simulations identified bottlenecks where fencing ended and caused a spillover effect, but while this was posited in the methods, I could not find anywhere in the Results or Discussion that backed up this conclusion. Suggest either making this result very clear in the manuscript, or deleting this in the abstract since it seems to not be supported by the paper.
Intro – I liked the Intro and think it does a good job of laying out the problem. One clarification is line 45: You state “Planning of roads in forested areas is a problem”. That seems to be the crux of your paper, since all of your modeling is taking place in forested habitat, and thus I expected this statement to be well fleshed out. However, you never tell us why planning roads in forests is a problem. I think this statement needs to be expanded on and explained to make a case for what the problem is and how the work you’re doing is related to it.
Methods – My primary concern with the paper is that lots of information is presented, and much of it seems to be thrown in whenever the authors come to it, without any explanation or road-map of where the methods are going. I think this would be a much better paper if some thought was put into guiding the reader better through what is going on. Consider whether you’re talking about something or using a term that hasn’t been introduced before, and about the order in which you’re explaining things; try to state up front what you’re doing and why, and then get more into the specifics.
Line 121: I would like more information about the digital forestry maps that were used, since the entire model and results are based on the accuracy of this layer. Has this been peer-reviewed, or has any ground-truthing been conducted? Can you offer the reader any reason to believe that this map was well constructed and does a good job of mapping the various forest composition types? Additionally, it seems that this map was used categorically in the model to determine the presence of shrubs and forest of a certain species type and age, etc. What categories of forest composition does this map contain? For instance, is there a category of spruce/fir over 20 years old, as is used in S2, or is that something you had to infer?
Line 161: The way that the sample sizes and years of data collection for moose vehicle collisions is presented is confusing. It would be helpful for the reader if the authors spent a paragraph in the Methods detailing when the MVC’s were collected, by who and therefore how we know if the spatial resolution is any good or consistent across the sample, and how the sample of MVCs is split and used in the later analysis. Please tell us why you used a sample size of 399 from 2002- to 2014 to calculate median distances, but seemingly a different sample of 59 locations from different years to test your model, and a sample of 78 on line 283 for different testing. And tell us this up front and in one paragraph, so we don’t have to go hunting for it throughout the paper to understand what you did.
Line 164: You found the median distance to buildings was 316m and then seemed to use this distance to make inferences about habitat loss adjacent to roads. Isn’t this forcing the assumption that moose respond the same to buildings and roads? Why couldn’t you use these 399 locations to actually calculate median distances to roads, and use that in the equation on line 164?
Line 171: The information about the topographic maps used in this analysis should also be presented in the text, not just in the figure caption, and the map source should be cited.
Line 172: You start this sentence “Our analysis based on all moose mortality locations showed that they seldom occurred in areas of open fields”. What analysis? Until now, you’ve never mentioned an analysis for open fields, and this is the result of such an analysis, so shouldn’t it go in the Results section? If this is related to the paragraph in Line 161, you need to tell us that you measured the distance from MVC to houses, buildings, open fields, and forest cover. This just presents the results (which should be in the results section, by the way) and forces the reader to guess what you did. It also seems like this paragraph and the paragraph starting with Line 161 are setting up the construction of what you later call the “realized moose habitat map”. If this is the case, that needs to be made clear here, because otherwise it’s unclear why you’re doing any of these distance based analyses.
Line 180: What is a district? How big is it, how many are in your study area? It makes reading your paper very difficult when you make your readers guess these things. Similarly, where did your density estimates come from, who collected them and how? Are they actual counts or estimates or an index, and was imperfect detection considered? What does ‘standardized’ mean here?
Line 182: Why do you now have 59 MVC locations, when on line 161 you used 399?
Line 184: How were these randomly generated road locations created? Within a certain boundary or what?
Line 190: Explain where the value of 1.414 comes from if crossing a cell diagonally.
Line 203: Where did the numbers that you applied to your HSI values in equations 3-5 come from? They seem very specific and right now they seem without any support as to why they were chosen.
Line 212: How were the known barriers incorporated into the friction surfaces? Did they produce an uncrossable barrier, or did they just make the cost to cross higher? And were all barriers treated the same?
Line 229: What moose density does an HSI value of 0.6 reflect? Does this split, for instance, mean no moose/some moose, or low moose/lots of moose, or something else?
Line 231: You don’t seem to have mentioned this interest of modeling mostly long range movements before this paragraph. If this was a goal of the paper, it should be introduced in the Introduction, and explained why this type of movement was being focused on. Also, please state whether the 6km distance was a biologically relevant one, or one just selected randomly?
Line 234: This is the first place in the paper that the terms “potential” and “realized” habitat quality are used. What is the difference between these? It needs to be made clear much earlier in the paper that you’re producing 2 habitat quality maps, and these terms should be clearly explained and assigned to each one.
Line 240: What is “hiding cover”? How was this defined/mapped?
Line 258: What is the reasoning behind having a cumulative rank based on these three combined ranks? Could you explain the purpose behind including the potential and realized and a 500m buffer, and not just picking one rank, or keeping them separate?
Line 267 and 280: What is the difference between these two testing sections, why is one called ‘model testing’ and one is ‘prognostic value’? Aren’t you testing the model predictive power in each one? Does one tell us something that the other one doesn’t? What is the point of the clusters? Please make it clearer in the text why both of these methods are necessary and what different information they provide.
Line 290 and 297: You got your acronym wrong, in both places you use ‘HIS’ instead of ‘HSI’. Please proofread everything carefully before sending to publication.
Line 301: The methods do not mention two separate simulations, this needs to be clarified. And if this is referring to the ‘potential’ and ‘realized’ habitat, those terms should be used here to make that clear. Also, is this referring to the friction grid that included known barriers, mentioned in line 212? If so, it also mentioned built up areas and industrial sites as well as underpasses, while this section only mentions “all currently-built fencing…”. Are these the same?
Line 303-308: This section seems like it should be in the Methods. And is “current fencing” and “no fencing” the same as “potential” and “realized”? If so, suggest picking one set of terms and sticking with it.
Figure 5: It’s unclear what the numbers on the x-axis are referring to.
Figure 6: This caption says the conditions were based on a 1km buffer on each side of the highway, while in the text you say 500m on each side.
Line 339: You say “…MVCs to high frequency crossing zones…”, but shouldn’t this be ‘all highway crossing zones’, not high frequency?
Line 343: It’s still not clear why you only used data from 2015-2017 to do this cluster test.
Line 353: It would be an interesting Discussion point to talk about why your cluster analysis differed so much between A1 and A2, and if the model for A2 seemed less successful since the cluster was 5km from the segment, and the MVCs are in all crossing zones (line 339), not high frequency like A1.
Line 370: It’s not until this line in the Discussion that you explain potential and realized. This needs to happen much earlier in the paper.
Discussion: I would suggest a complete rewrite of this entire section. Currently, your Discussion focuses on ideas that seem not very relevant to your analysis, such as topography, agriculture, the sub-sample of successful crossings, and bias, and it doesn’t even develop any of these ideas thoroughly. I would recommend a first paragraph that focuses much more on the key findings and take-home messages from your results, such as whether you think your models are successful, whether this means crossings can be predicted by habitat, and whether the crossings modeled from only habitat differ from those modeled when fencing is considered. Keep it focused on your work and not bogged down with mentions of other papers. Then in the next paragraphs you can discuss interesting findings and compare your results with the literature. Some things that would be good to have covered in the discussion seem to be: Was the model equally successful for A1 versus A2? It seems like from your cluster results that it wasn’t, since A2 only had one crossing 5km away, or that the crossings differed on these two highways. How do your crossing predictions relate to the existing overpasses, fences, and bridges that were already there? Are there any specific management recommendations that you would like to make for people wanting to add additional crossing structures or fences, particularly with regard to how close roads are to forest cover or buildings? And it would be really interesting to know how your predictions differed from your ‘potential’, ‘realized’, and 500m buffer models. Each of these ideas in a separate paragraph, developed with comparisons to other studies and considered in a management framework, I think would make a more relevant Discussion.
Author Response
Comments of the Rev#1
Comment: Abstract Line 29: You state that simulations identified bottlenecks where fencing ended and caused a spillover effect, but while this was posited in the methods, I could not find anywhere in the Results or Discussion that backed up this conclusion. Suggest either making this result very clear in the manuscript, or deleting this in the abstract since it seems to not be supported by the paper.
Answer: we add text (Lines 312-313), listing actual spillover places “Spillovers were identified at 38, 54, 78 nd 138 km on the highway A1, and 22, 91 and 114 km on A2. Actual MVC data for 2002–2014 confirm presence of spillover effect.“
Comment: Intro – I liked the Intro and think it does a good job of laying out the problem. One clarification is line 45: You state “Planning of roads in forested areas is a problem”. That seems to be the crux of your paper, since all of your modeling is taking place in forested habitat, and thus I expected this statement to be well fleshed out. However, you never tell us why planning roads in forests is a problem. I think this statement needs to be expanded on and explained to make a case for what the problem is and how the work you’re doing is related to it.
Answer: we added text: “Forests are primary habitats for many wild ungulate species, in the case of Lithuania it is moose (Alces alces Linnaeus, 1758), red deer (Cervus elaphus Linnaeus, 1758), roe deer (Capreolus capreolus Linnaeus, 1758), wild boar (Sus scrofa Linnaeus, 1758). In the last years, collisions with European bison (Bison bonasus Linnaeus, 1758) and fallow deer (Dama dama Linnaeus, 1758) took place; both these species are also forest animals. Moreover, two times per year moose are moving between winter grounds and summer areas, this increasing numbers of moose-vehicle collisions (MVC) in the forested areas.“ This should be an explanation about importance of the forest habitat in relation to transport collisions with ungulates.
Comment: Methods – My primary concern with the paper is that lots of information is presented, and much of it seems to be thrown in whenever the authors come to it, without any explanation or road-map of where the methods are going. I think this would be a much better paper if some thought was put into guiding the reader better through what is going on. Consider whether you’re talking about something or using a term that hasn’t been introduced before, and about the order in which you’re explaining things; try to state up front what you’re doing and why, and then get more into the specifics.
Answer: to answer Rev#1 concerns, we introduced first subchapter in Methods, 2.3 (after 2.1 – study site, and 2.2 – MVC data) – workflow, where we explain an order things were done, from habitat map to model testing.
Comment: Line 121: I would like more information about the digital forestry maps that were used, since the entire model and results are based on the accuracy of this layer. Has this been peer-reviewed, or has any ground-truthing been conducted? Can you offer the reader any reason to believe that this map was well constructed and does a good job of mapping the various forest composition types? Additionally, it seems that this map was used categorically in the model to determine the presence of shrubs and forest of a certain species type and age, etc. What categories of forest composition does this map contain? For instance, is there a category of spruce/fir over 20 years old, as is used in S2, or is that something you had to infer?
Answer: answer is positive, we used map of the digitized forest cadastre. No inferring was needed, all information we need was available. Such maps exist in Lithuania for a long time, earlier they were in the paper format. Ground truth was made at the state level, throughout all forest territory of Lithuania, and is constantly re-newed. At the parcel (smallest uniform forest area) level, information on the forest composition and age is present, listing dominant and other tree species, as well as other habitats (shrub, wetland, open area, clearcut, etc.) used to create HIS model. We are sorry about supposing such information should be understandable for the reader automatically. Explanation is added to the text at the Line 122 of original text.
The only problem in 2014, when modeling was made, was availability of these maps – or, rather, a cost of data availability. We used license of the Nature Research Centre (where LB was working, and AK is Phd student now) for this map.
Comment: Line 161: The way that the sample sizes and years of data collection for moose vehicle collisions is presented is confusing. It would be helpful for the reader if the authors spent a paragraph in the Methods detailing when the MVC’s were collected, by who and therefore how we know if the spatial resolution is any good or consistent across the sample, and how the sample of MVCs is split and used in the later analysis. Please tell us why you used a sample size of 399 from 2002- to 2014 to calculate median distances, but seemingly a different sample of 59 locations from different years to test your model, and a sample of 78 on line 283 for different testing. And tell us this up front and in one paragraph, so we don’t have to go hunting for it throughout the paper to understand what you did.
Answer: Main MVC data source for this study were the Lithuanian Police Traffic Supervision Service, recording animal-vehicle collisions and maintaining database starting from 2002. From 2007, Nature Research Centre (LB group) is doing road ecology research, collecting data on all roadkilled animals (over 300 000 km of designated driving in the country). As moose is really big animal, nearly 100% of roadkill cases are reported to Traffic Supervision Service (100% on the highways A1 and A2, due to high speed allowed and severity of crashes). Location of MVC is to nearest 100 m marking of the road or GPS coordinate. We converted these data to GIS layer (Kučas, Balčiauskas, 2019 after reviews).
Model was created in 2014. On the highways A1 and A2 sample size was 399 MVC in 2002-2014, 59 MVC in 2010-2013 used for testing (this is a subsample) and new data on 78 MVC in 2015-2017, used to see if model still works. We introduced 2.2 chapter to describe samples, and made correction of the text below.
Comment: Line 164: You found the median distance to buildings was 316m and then seemed to use this distance to make inferences about habitat loss adjacent to roads. Isn’t this forcing the assumption that moose respond the same to buildings and roads? Why couldn’t you use these 399 locations to actually calculate median distances to roads, and use that in the equation on line 164?
Answer: mistype on the line 166, instead of roads there should be “built footprint”. MVC were on the roads A1 and A2, so there is no distance, or median is zero.
Comment: Line 171: The information about the topographic maps used in this analysis should also be presented in the text, not just in the figure caption, and the map source should be cited.
Answer: Digital 1:10,000 topographic maps of the country now are available at (https://www.geoportal.lt/geoportal/nacionaline-zemes-tarnyba-prie-zemes-ukio-ministerijos) – service paid, Nature research Centre has the license. We added information to the text, former Lines 161-162.
Comment: Line 172: You start this sentence “Our analysis based on all moose mortality locations showed that they seldom occurred in areas of open fields”. What analysis? Until now, you’ve never mentioned an analysis for open fields, and this is the result of such an analysis, so shouldn’t it go in the Results section? If this is related to the paragraph in Line 161, you need to tell us that you measured the distance from MVC to houses, buildings, open fields, and forest cover. This just presents the results (which should be in the results section, by the way) and forces the reader to guess what you did. It also seems like this paragraph and the paragraph starting with Line 161 are setting up the construction of what you later call the “realized moose habitat map”. If this is the case, that needs to be made clear here, because otherwise it’s unclear why you’re doing any of these distance based analyses.
Answer: we partially rewrote this text (Lines 162-172) to make clear, that open areas are meant as having no hiding cover. Analysis was testing distances between MVC locations and nearest forest.
However, we strongly ask text remain in place, as analyses were done to see, if we need to adjust Allen’s Model II – as a result, adjustment was done as for distance from the built footprint, but not open areas. So, despite Figure 2 is a result, it has methodological purpose.
Comment: Line 180: What is a district? How big is it, how many are in your study area? It makes reading your paper very difficult when you make your readers guess these things. Similarly, where did your density estimates come from, who collected them and how? Are they actual counts or estimates or an index, and was imperfect detection considered? What does ‘standardized’ mean here?
Answer: district is unit of administrative division of the country, to be easier understood by foreign reader, we changed “district” to “municipality”. Lithuania has got 60 national municipalities (https://osp.stat.gov.lt/regionine-statistika-pagal-statistikos-sritis) that range from 440 to 2216 km2 (mean = 1282.2 km2). 23 municipalities were fully or partially covered by 30 km buffer zone around highways A1 and A1 and A2.
Density estimates of moose comes from the official source, Ministry of Environment of the Republic of Lithuania (so far not available online). Estimate is the number of individuals. These numbers are checked via ungulate monitoring, carried out by Nature Research Centre, and generally good agreement for moose was found in 2008.
Standardization here was intended mean normalization of the values to fit into the range between 0 and 1. We used formula: (value-min)/(max-min).
We made changes to the former Lines 179-185, expanding text to incorporate details shown above.
Comment: Line 182: Why do you now have 59 MVC locations, when on line 161 you used 399?
Answer: because of different time span, n = 399 is number of MVC in 2002-2014, and n = 59 is number of MVC in 2009-2013 (subsample for testing). Time periods now are clearly shown in the text.
Comment: Line 184: How were these randomly generated road locations created? Within a certain boundary or what?
Answer: Randomly generated locations (N = 59) on the highways A1 and A2 mean dotted locations on the road area, comparable to dotted location of MVC.
Comment: Line 190: Explain where the value of 1.414 comes from if crossing a cell diagonally.
Answer: for a square cell with length of sides = 1, diagonal is SQRT (2) = 1.414…
Comment: Comment: Line 203: Where did the numbers that you applied to your HSI values in equations 3-5 come from? They seem very specific and right now they seem without any support as to why they were chosen.
Answer: Formulae were derived empirically, to best ensure that the generated pathways followed high quality habitat patches. Thus the resulting trajectories cross the areas of highest theoretical forage potential.
Comment: Line 212: How were the known barriers incorporated into the friction surfaces? Did they produce an uncrossable barrier, or did they just make the cost to cross higher? And were all barriers treated the same?
Answer: such barriers as fences and continuously built areas were treated as uncrossable (e). However, fences change direction of the movement, forcing moose to move along the road/fence, looking for underpasses or other possibility to cross (gaps, gates, crossroads) – this is how spillover effect is created.
Comment: Line 229: What moose density does an HSI value of 0.6 reflect? Does this split, for instance, mean no moose/some moose, or low moose/lots of moose, or something else?
Answer: HIS > 0.6 reflect best moose habitat. We mean, that not a moose number or a habitat was stratified – we stratified number of entry/exit points according actual moose densities in municipalities (more moose – higher number of entry/exit points assigned). We rewrote text to make this clear.
Comment: Line 231: You don’t seem to have mentioned this interest of modeling mostly long range movements before this paragraph. If this was a goal of the paper, it should be introduced in the Introduction, and explained why this type of movement was being focused on. Also, please state whether the 6km distance was a biologically relevant one, or one just selected randomly?
Answer: on the Lines 75-76 we stated, that “Described method of mapping moose movement did not seek to mimic individual moose routes across a landscape, often driven by stochastic events that are impossible to express quantitatively.” Answering comment above, we added “long-range” movement characterization to several manuscript parts, starting from Abstract, then to Line 76, etc.
6 km is relevant to moose biology. It is known (Allen et al., 1987), that minimum 9000 ha can support entire moose populations on a reasonably long-term basis; we reference this (Line 151-152). Radius of this area is ca. 5.4 km, so 6 km is just on the safe side, as 9000 ha is minimum area. We added text to Line 234.
Comment: Line 234: This is the first place in the paper that the terms “potential” and “realized” habitat quality are used. What is the difference between these? It needs to be made clear much earlier in the paper that you’re producing 2 habitat quality maps, and these terms should be clearly explained and assigned to each one.
Answer: realized habitat includes all limitations, imposed by human activities (such as built up footprint, industrial areas etc.). however, Line 234 and close text did not refer to “potential” and “realized” habitat quality. So, we explained issue on Line 259.
Comment: Line 240: What is “hiding cover”? How was this defined/mapped?
Answer: hiding cover is the same as forest cover. To avoid confusion, former term was changed throughout, starting from the Line 174.
Comment: Line 258: What is the reasoning behind having a cumulative rank based on these three combined ranks? Could you explain the purpose behind including the potential and realized and a 500m buffer, and not just picking one rank, or keeping them separate?
Answer: Reason of having cumulative rank (Lines 258-261) is seeking of better evaluation of differences of the natural habitats and human influence. Particularly, sites may have excellent potential moose habitat, but poor realized habitat due to proximity of the built up areas (so Rank 1 and Rank 2 were computed accordingly); habitat quality in closest buffer to highway may differ from that in distance (so Rank 3 was computed to reflect it in both sides of the road). Averaging all three ranks reflects all three habitat types and their spatial variation.
Line 267 and 280: What is the difference between these two testing sections, why is one called ‘model testing’ and one is ‘prognostic value’? Aren’t you testing the model predictive power in each one? Does one tell us something that the other one doesn’t? What is the point of the clusters? Please make it clearer in the text why both of these methods are necessary and what different information they provide.
Answer: after the changes, 2.8 is Moose movement model testing, and 2.9 is Prognostic value of the moose movement model. In the first case we test, if obtained model is valid (if actual MVC are correlated to the road crossing zones in the model). For the testing we used subsample of all MVC (59 MVC, occured in 2009-2013).
Second test shows predictive power of the model – we tested MVC which occurred after the model was made. So, the second testing answer a question – if moose movement model can have value in the future years (three years in our case). Answer is important, so we tested not only distance between actual MCV and predicted crossing zones, but also distance from predicted crossing zones and clusters of MVC. Clusters are the essence of rodkills, as these are short road segments, where MVC are concentrated.
Comment: Line 290 and 297: You got your acronym wrong, in both places you use ‘HIS’ instead of ‘HSI’. Please proofread everything carefully before sending to publication.
Answer: we apologize mistypes, it is autocorrecting feature of the Word involved; manuscript was re-checked and changes made.
Comment: Line 301: The methods do not mention two separate simulations, this needs to be clarified. And if this is referring to the ‘potential’ and ‘realized’ habitat, those terms should be used here to make that clear. Also, is this referring to the friction grid that included known barriers, mentioned in line 212? If so, it also mentioned built up areas and industrial sites as well as underpasses, while this section only mentions “all currently-built fencing…”. Are these the same?
Answer: We are most thankful for this comment, as it helped to make text more understandable. We did moose movement simulation twice.
In the first, we assumed no wildlife fences are present, and generated distinct moose movement pathways based just on the realized habitat. Realized habitat has a limitation of built-up footprint imposed, but not the fences.
For the second simulation, also based on the realized habitat, all wildlife fences were treated as impermeable barrier, thus limiting number of the pathways. At Results we added numbers of road segments with moose crossings according the both simulations.
Comment: Line 303-308: This section seems like it should be in the Methods. And is “current fencing” and “no fencing” the same as “potential” and “realized”? If so, suggest picking one set of terms and sticking with it.
Answer: No, terms “potential” and “realized” refer to HSI map, where realized habitat incorporates restrictions to a habitat suitability, imposed by built-up footprint. Terms “current fencing” and “no fencing” are related to the simulation of moose movement, both based on the realized habitat. We made change to the text, and partly moved it to Methods. For easier understanding, explanation is clarified in 2.3 Workflow section.
Comment: Figure 5: It’s unclear what the numbers on the x-axis are referring to.
Answer: it is kilometer marks on the highways A1 and A2 (as is written)
Comment: Figure 6: This caption says the conditions were based on a 1 km buffer on each side of the highway, while in the text you say 500m on each side.
Answer: it was mistake in the caption, it now sounds “within a 1 km buffer including both sides of the highway“
Comment: Line 339: You say “…MVCs to high frequency crossing zones…”, but shouldn’t this be ‘all highway crossing zones’, not high frequency?
Answer: no, text is correct, for highway A1 we mean zones with high frequency of modeled moose crossings, and for A2 – all crossing zones.
Comment: Line 343: It’s still not clear why you only used data from 2015-2017 to do this cluster test.
Answer: it was because at the moment of writing, additional data of moose-vehicle collisions were available for 2015–2017 years (data for 2018 are still not processed). “prognostic” means, that we were testing time period after the model development. Idea was – if model is not working after the several years, then his validity is low. Clusters are the essential localities of MVC, most related to traffic security.
Comment: Line 353: It would be an interesting Discussion point to talk about why your cluster analysis differed so much between A1 and A2, and if the model for A2 seemed less successful since the cluster was 5km from the segment, and the MVCs are in all crossing zones (line 339), not high frequency like A1.
Answer: Cluster on A2 being 5 km from the predicted segment depends on the wildlife fencing, implemented after 2014 and thus not accounted by the model.
One of the causes of differences in MVC cluster formations on A1 and A2 is related to fencing (see Figure 1B in the manuscript). However, cluster analysis is out of the scope of current manuscript. As our publication of the ungulate clusters is already reviewer in another journal, we are restricted to add text in Results, so just moderate change to Discussion and Conclusions was introduced.
Comment: Line 370: It’s not until this line in the Discussion that you explain potential and realized. This needs to happen much earlier in the paper.
Answer: terms explained in the introduced 2.3. chapter on the workflow.
Comment: Discussion: I would suggest a complete rewrite of this entire section. Currently, your Discussion focuses on ideas that seem not very relevant to your analysis, such as topography, agriculture, the sub-sample of successful crossings, and bias, and it doesn’t even develop any of these ideas thoroughly. I would recommend a first paragraph that focuses much more on the key findings and take-home messages from your results, such as whether you think your models are successful, whether this means crossings can be predicted by habitat, and whether the crossings modeled from only habitat differ from those modeled when fencing is considered. Keep it focused on your work and not bogged down with mentions of other papers. Then in the next paragraphs you can discuss interesting findings and compare your results with the literature. Some things that would be good to have covered in the discussion seem to be: Was the model equally successful for A1 versus A2? It seems like from your cluster results that it wasn’t, since A2 only had one crossing 5km away, or that the crossings differed on these two highways. How do your crossing predictions relate to the existing overpasses, fences, and bridges that were already there? Are there any specific management recommendations that you would like to make for people wanting to add additional crossing structures or fences, particularly with regard to how close roads are to forest cover or buildings? And it would be really interesting to know how your predictions differed from your ‘potential’, ‘realized’, and 500m buffer models. Each of these ideas in a separate paragraph, developed with comparisons to other studies and considered in a management framework, I think would make a more relevant Discussion.
Answer: we did as recommended, rewriting Discussion in the way Rev#1 asked.

Reviewer 2 Report
This manuscript uses a model to investigate the least-cost movement path used by moose. It models the probability of places of road- crossing. This work is important and contributes to recent and increasing discussions about the effect of highways on animal movement. Below are my comments per line:
Line 45. Explain the problems of planning roads in forested areas. Alternatively, mention the problems suggested by the reference you are using.
Line 49. All the sources here are MVC per year, except the data from Maine. So try to be consistent or explain the importance of citing a source that shows different time/scale.
Line 54. I suggest the authors stopping at the populations, the piece “along major highways” without discussion does not add to the manuscript.
Line 70. The authors need to explain what the least-cost path models showed for carnivore species and the wildlife they are citing. After the references explain what the models found.
Line 72. I suggest the authors to discuss recommendations instead of providing.
Line 74 to 78. The font type and/or size is different from the rest of the text. Be consistent.
Line 88. Do the authors have a citation for how the moose migrations across the highways were affected? Alternatively, explain how. It is unclear if the authors are talking about car accidents or other issues.
Fig.1. The map needs improvements. Which one is 1a or 1b?
Line 109. Here is not the place to suggest the model.
The method should describe why the authors are using the same model. So, I suggest the authors to re-write the paragraph.
Line 115. Instead, to mention that Allen’s model does not incorporate topography, especially if the authors applied the same model. If there is something to say about topography, include it in the discussion. This information here may cause more problems than help the authors.
Line 139. Repeat what S1 to S4 represents here again. It is difficult to go back and read all the bullet points, so make it easier for the reader. Or better, move the equation to the top of the model assumptions. Line 138 should go above Line 122.
Line 142 to Line 148. This paragraph is difficult to parse. Could the authors re-explain and mention why the method is considered restrictive?
Line 172. This paragraph should be in the results. I suggest starting this paragraph by explaining the uncertainties and not including agriculture, and move the beginning of the paragraph to the second part.
Line 196 to 199. I am not sure why it would be safe to assume that the Model II reflects the moose forage. It will make the paper stronger if the authors report that one of four variables was removed because it did not relate to moose feeding and explain why it did not relate.
Line 238. Should it be a period after characterized?
Line 280 to 285. The session should be in results.
Line 298. I think it is better to say that there was a statistically significant difference…
Line 269. In the first line of the discussion, it is better to restate if you found what you were looking for or not. Also, evaluate the model.
Line 378. What do the authors mean by “stationary” moose? Are these mooses that do not migrate? So, is there a better term, like Non-migratory moose. I suggest using non-migratory, which sounds better.
Line 379. This phrase should be in results.
Line 388. This paragraph is the same one in line 293.
Line 412. The authors mention that there is no mimic of individual moose, but if I am not wrong in the introduction, they say individual moose were considered as well.
Line 412 to 420. The font is different.
Author Response
Comments of the Rev#2
Comment: Line 45. Explain the problems of planning roads in forested areas. Alternatively, mention the problems suggested by the reference you are using.
Answer: the following text was added “Planning of roads in forested areas is a problem [13]. Forests are primary habitats for many wild ungulate species, in the case of Lithuania it is moose (Alces alces Linnaeus, 1758), red deer (Cervus elaphus Linnaeus, 1758), roe deer (Capreolus capreolus Linnaeus, 1758), wild boar (Sus scrofa Linnaeus, 1758). In the last years, collisions with European bison (Bison bonasus Linnaeus, 1758) and fallow deer (Dama dama Linnaeus, 1758) took place; both these species are also forest animals. Moreover, two times per year moose are moving between winter grounds and summer areas, this increasing numbers of moose-vehicle collisions (MVC) in the forested areas.“
Comment: Line 49. All the sources here are MVC per year, except the data from Maine. So try to be consistent or explain the importance of citing a source that shows different time/scale.
Answer: for the Maine we adder “with yearly average over 500 MVC“, to be consistent in time scale. We also added data for Lithuania: “In Lithuania, the number of MVC is rising every year, from 12 in 2003 till 203 in 2016, and is the main cause of human fatalities in vehicle-wildlife collisions [20]“.
Comment: Line 54. I suggest the authors stopping at the populations, the piece “along major highways” without discussion does not add to the manuscript.
Answer: we agree, recommended change done.
Comment: Line 70. The authors need to explain what the least-cost path models showed for carnivore species and the wildlife they are citing. After the references explain what the models found.
Answer: we added text “For the fisher (Martes pennanti Erxleben, 1777) and bobcat (Lynx rufus Schreber, 1777) least-cost model wildlife corridors identified better, than for American black bear (Ursus americanus Pallas, 1780) [38]. However, for African forest elephant (Loxodonta africana Blumenbach, 1797), forest buffalo (Syncerus caffer nanus Boddaert, 1785), western lowland gorilla (Gorilla gorilla gorilla Savage and Wyman, 1847), and central chimpanzee (Pan troglodytes troglodytes Blumenbach, 1775) movement corridors with the least-cost model overlapped only partially [40].“
Comment: Line 72. I suggest the authors to discuss recommendations instead of providing.
Answer: we agree, recommended change done.
Comment: Line 74 to 78. The font type and/or size is different from the rest of the text. Be consistent.
Answer: we checked original manuscript, and the font was not different, so we changed it in the current version. Possibly it is the loss of formatting, we apologize.
Comment: Line 88. Do the authors have a citation for how the moose migrations across the highways were affected? Alternatively, explain how. It is unclear if the authors are talking about car accidents or other issues.
Answer: text changed to give explanation “Moose migrations across the highways A1 and A2 were affected by the wildlife fencing (forcing moose to move along the fences), where just underpasses and river bridges with sufficient dimensions allowed crossing“. We are talking about the possibility to cross, not about MVC.
Comment: Fig.1. The map needs improvements. Which one is 1a or 1b?
Answer: we put letters A and B to the Figure 1.
Comment: Line 109. Here is not the place to suggest the model. The method should describe why the authors are using the same model. So, I suggest the authors to re-write the paragraph.
Answer: we changed text according suggestion.
Comment: Line 115. Instead, to mention that Allen’s model does not incorporate topography, especially if the authors applied the same model. If there is something to say about topography, include it in the discussion. This information here may cause more problems than help the authors.
Answer: we changed text according suggestion.
Comment: Line 139. Repeat what S1 to S4 represents here again. It is difficult to go back and read all the bullet points, so make it easier for the reader. Or better, move the equation to the top of the model assumptions. Line 138 should go above Line 122.
Answer: We think repeating large piece of text is not acceptable, therefore, equation was moved up in the text, as advised.
Comment: Line 142 to Line 148. This paragraph is difficult to parse. Could the authors reexplain and mention why the method is considered restrictive?
Answer: we rewrote tis paragraph as “We, however, are using a quantitative habitat map as the basis for performing moose movement simulations, not trying to assess the moose carrying capacity of the study area. In this context method of calculating the composite HSI index, presented in (1), is far too restrictive. We presume that the absence of any of the model components (S1 to S4) in an evaluation area ought not to reduce its value to zero, but reduce it proportionally. We decided to use an arithmetic mean to compute the composite index and believe that our less restrictive equation better represents the value of the evaluation areas of migrating or foraging moose:“.
Comment: Line 172. This paragraph should be in the results. I suggest starting this paragraph by explaining the uncertainties and not including agriculture, and move the beginning of the paragraph to the second part.
Answer: we very well understand this comment, however, paragraphs starting at Line 162 and second one, at Line 172 we designed to show, that close to built-up areas moose habitat quality decreases, so there are less MVC cases (and this was used computing HSI model), while closiness to agricultural areas did not influence locations of MVC. This is not the firm rebuttal, so we re-arranged text for clarity, but did not move it to Results.
Comment: Line 196 to 199. I am not sure why it would be safe to assume that the Model II reflects the moose forage. It will make the paper stronger if the authors report that one of four variables was removed because it did not relate to moose feeding and explain why it did not relate.
Answer: none of the S1–S4 variables were removed, just S3 provides not only food, but also is significant for the cover. We removed conflicting sentence.
Comment: Line 238. Should it be a period after characterized?
Answer: Yes, and we apologize mistype.
Comment: Line 280 to 285. The session should be in results.
Rebuttal: we think these lines characterize Material.
Comment: Line 298. I think it is better to say that there was a statistically significant difference…
Answer: we changed the text “Distribution of distances between MVCs and sites with above HSI values and between MVCs and random locations was different (Figure S2), and the median distances, being 403 m and 1344 m, respectively, differed significantly (Wilcoxon W = 936, P < 0.0001).“
Comment: Line 269. In the first line of the discussion, it is better to restate if you found what you were looking for or not. Also, evaluate the model.
Answer: it is Line 369 most possibly. We acknowledge this comment, as it goes in line with Rev#1, and Discussion was rewritten and restructured.
Comment: Line 378. What do the authors mean by “stationary” moose? Are these mooses that do not migrate? So, is there a better term, like Non-migratory moose. I suggest using non-migratory, which sounds better.
Answer: we accepted proposed term “non-migratory”, changes done in the text.
Comment: Line 379. This phrase should be in results.
Answer: do not agree, or Line number is mistakenly shown.
Comment: Line 388. This paragraph is the same one in line 293.
Answer: we changed text in the Discussion, after Line 388. “Despite generally good agreement between average HSI values and 5-year moose density averages in the districts (or their parts) within the 30 km buffer on each side of both highways, in the four districts the model overestimated moose habitat potential. In these four districts, however, a relatively high number of MVCs were reported [62]. Given that moose density (or moose evidence) and habitat quality is correlated with MVC locations [16,63–65], moose census numbers for these districts were likely underestimated.“
Comment: Line 412. The authors mention that there is no mimic of individual moose, but if I am not wrong in the introduction, they say individual moose were considered as well.
Answer: in the Line 153 we say “We were interested in doing the evaluation at the level of an individual moose.“ However, context here is an area, or home range, not movements. Thus, we thisk, no changes are needed in Line 412.
Comment: Line 412 to 420. The font is different.
Answer: we checked original manuscript, and the font was not different, so we changed it in the current version. Possibly it is the loss of formatting, we apologize.
